# Effect of Permissive Underfeeding with Intensive Insulin Therapy on MCP-1, sICAM-1, and TF in Critically Ill Patients

**DOI:** 10.3390/nu11050987

**Published:** 2019-04-30

**Authors:** Ahmad Aljada, Ghada Fahad AlGwaiz, Demah AlAyadhi, Emad Masuadi, Mahmoud Zahra, Shahad H. Al-Matar, Ahmad Al-Bawab, Waleed Tamimi, Dunia Jawdat, Abdulaziz Al-Dawood, Maram H. Sakkijha, Musharaf Sadat, Yaseen M. Arabi

**Affiliations:** 1Department of Biochemistry and Molecular Medicine, College of Medicine, Alfaisal University, Riyadh 11533, Saudi Arabia; aaljada@alfaisal.edu; 2Department of Basic Medical Sciences, King Saud bin Abdulaziz University for Health Sciences, King Abdul Aziz Medical City, Ministry of National Guard—Health Affairs, Riyadh 11481, Saudi Arabia; Ghada_fahad@live.com (G.F.A.); iyadhi515@ksau-hs.edu.sa (D.A.); mahmoud.mzhra@gmail.com (M.Z.); Matar028@ksau-hs.edu.sa (S.H.A.-M.); bawaba@ksau-hs.edu.sa (A.A.-B.); 3Research Unit, Department of Medical Education, College of Medicine, King Saud bin Abdulaziz University for Health Sciences, King Abdul Aziz Medical City, Ministry of National Guard—Health Affairs, Riyadh 11481, Saudi Arabia; Masuadie@ksau-hs.edu.sa; 4Department of Clinical Laboratory, King Saud bin Abdulaziz University for Health Sciences, King Abdullah International Medical Research Center, King Abdulaziz Medical City, Riyadh 11481, Saudi Arabia; TamimiW@NGHA.MED.SA; 5Cord Blood Bank, King Saud bin Abdulaziz University for Health Sciences, King Abdullah International Medical Research Center, King Abdulaziz Medical City, Riyadh 11481, Saudi Arabia; JawdatD@NGHA.MED.SA; 6College of Medicine, King Saud bin Abdulaziz University for Health Sciences, King Abdullah International Medical Research Center, Intensive Care Department, King Abdulaziz Medical City, Riyadh 11481, Saudi Arabia; dawooda@ngha.med.sa (A.A.-D.); SakkijhaM@NGHA.MED.SA (M.H.S.); sadatmu@ngha.med.sa (M.S.)

**Keywords:** permissive underfeeding, insulin infusion, caloric intake, inflammation, MCP-1, sICAM-1 and tissue factor

## Abstract

Purpose: This study examined the effect of permissive underfeeding compared to target feeding and intensive insulin therapy (IIT) compared to conventional insulin therapy (CIT) on the inflammatory mediators monocyte chemoattractant protein 1 (MCP-1), soluble intercellular adhesion molecule 1 (sICAM-1), and tissue factor (TF) in critically ill patients. Methodology: This was a substudy of a 2 × 2 factorial design randomized controlled trial in which intensive care unit (ICU) patients were randomized into permissive underfeeding compared to target feeding groups and into IIT compared to CIT groups (ISRCTN96294863). In this substudy, we included 91 patients with almost equal numbers across randomization groups. Blood samples were collected at baseline and at days 3, 5, and 7 of an ICU stay. Linear mixed models were used to assess the differences in MCP-1, sICAM-1, and TF across randomization groups over time. Results: Baseline characteristics were balanced across randomization groups. Daily caloric intake was significantly higher in the target feeding than in the permissive underfeeding groups (*P*-value < 0.01), and the daily insulin dose was significantly higher in the IIT than in the CIT groups (*P*-value < 0.01). MCP-1, sICAM-1, and TF did not show any significant difference between the randomization groups, while there was a time effect for MCP-1. Baseline sequential organ failure assessment (SOFA) score and platelets had a significant effect on sICAM-1 (*P*-value < 0.01). For TF, there was a significant association with age (*P*-value < 0.01). Conclusions: Although it has been previously demonstrated that insulin inhibits MCP-1, sICAM-1 in critically ill patients, and TF in non-critically ill patients, our study demonstrated that IIT in critically ill patients did not affect these inflammatory mediators. Similarly, caloric intake had a negligible effect on the inflammatory mediators studied.

## 1. Introduction

Global activation of the immune system through stress hyperglycemia is one of the critical illness characteristics resulting in a systemic inflammatory response, causing the release of numerous inflammatory mediators and cytokines [1]. Inflammation in critical illness has been extensively studied, and several inflammatory mediators have been measured in intensive care unit (ICU) patients, including C-reactive protein (CRP), procalcitonin (PCT), interleukin-6 (IL-6), and tumor necrosis factor-alpha (TNF-α) [2]. Insulin alleviates the detrimental effects of hyperglycemia through its metabolic regulation. It also modulates inflammatory mediators directly and enhances immunocompetence. Studies on inflammatory mediators such as MCP-1, sICAM-1, and TF, the primary initiator of the coagulation pathway, in relation to insulin therapy and glucose levels have shown a strong relation between the two variables. Hyperglycemia increased MCP-1, sICAM-1, and TF concentrations in healthy people and in patients with type 2 diabetes mellitus [3,4]. On the other hand, insulin was observed to suppress MCP-1, sICAM-1, and TF in normal obese subjects and obese patients with type 2 diabetes mellitus [4,5]. An insulin effect was also demonstrated on MCP-1 in critically ill patients with sepsis [6] and on sICAM-1 in 403 patients with prolonged critical illness [7].

Macronutrient intake has been demonstrated to exert a proinflammatory response. Proinflammatory transcription factors, including nuclear factor κB (NF-κB), activator protein-1 (AP-1), and early growth response 1 (Egr-1), were activated by glucose intake with an increase in the expression of TNFα, matrix metalloproteinases 2 (MMP-2), MMP-9, and TF [8]. The activation of NF-κB has also been shown following an intake of mixed meal [9]. Thus, the intake of glucose leads to inflammatory stress and a prothrombotic state. A large mixed meal induced a more intense and prolonged inflammatory stress response in the obese than in subjects of a normal weight [10]. Thus, hyperglycemia and an increased caloric intake may have a proinflammatory effect, whereas insulin could exert anti-inflammatory properties. The current study was a subset from a 2 × 2 factorial design randomized controlled trial conducted in a tertiary hospital to examine the effect of permissive underfeeding compared to target feeding and IIT compared to CIT on inflammatory mediators, MCP-1, sICAM-1, and TF, in critically ill patients. 

## 2. Methods and Reagents

### 2.1. Subjects

This was a substudy of a 2 × 2 factorial design randomized controlled trial conducted in a 21-bed medical–surgical ICU of a tertiary care academic hospital that evaluated the effects of permissive underfeeding compared to target feeding and intensive insulin therapy (IIT) compared to conventional insulin therapy (CIT) on the outcomes of 240 critically ill patients (ISRCTN96294863-http://www.controlled-trials.com/ISRCTN96294863) [11]. The study was approved by the Institutional Review Board (IRBC- 2005/030) of King Abdullah International Medical Research Center (KAIMRC). In this trial, patients aged >18 years with a blood glucose concentration of >6.1 mmol/L (110 mg/dL) receiving enteral feeding and expected to stay for 48 h were randomly assigned to permissive underfeeding or target feeding groups (caloric goal: 60%–70% compared to 90%–100% of calculated requirement, respectively) and to IIT or CIT groups (target blood glucose: 4.4–6.1 compared to 10–11.1 mmol/L, respectively). The study found no difference in the primary outcome of 28-day mortality, although permissive underfeeding was associated with a reduction in the secondary outcome of hospital mortality compared to target feeding (30.0% compared to 42.5%; relative risk: 0.71; 95% confidence interval (CI): 0.50, 0.99; *P* = 0.04), with no significant difference between IIT and CIT groups. In this a priori substudy, consecutive patients enrolled in the main trial between December 2006 and December 2007 and who were expected to stay at least 3 days in the ICU as judged by their treating physician consented to participate in this substudy. Blood samples were collected in Ethylenediaminetetraacetic acid (EDTA)- and citrate-treated tubes at baseline and on days 3, 5, and 7 of enrollment in the trial. The samples were immediately centrifuged at 4 °C for 20 min at 1600× *g*. Plasma samples were stored at −80 °C until assaying. Patients with ≥3 samples were included in the analyses. Registration: ISRCTN96294863-https://doi.org/10.1186/ISRCTN96294863. 

### 2.2. Interventions

#### 2.2.1. Permissive Underfeeding Compared to Target Feeding

Harris–Benedict equations were used for a standard caloric requirement estimation [12] targeting 90%–100% of the standard caloric requirement (target feeding group) and 60%–70% of the standard caloric requirement (permissive underfeeding group). Our published enteral feeding protocol was utilized for enteral feeding administration [13], and the type of formula used was left to the discretion of the treating team. Caloric intake was assessed on a daily basis by a dietitian, taking into consideration intravenous dextrose and propofol infusions. The protein requirement was calculated as 0.8–1.5 g/kg based on the patient’s condition and underlying diseases. Additional protein (Resource Beneprotein; Nestle Healthcare Nutrition Inc, Minneapolis, MN, USA) was added for the permissive underfeeding group to assure full protein requirements without affecting the assigned caloric intake and to avoid protein malnutrition. 

#### 2.2.2. IIT Compared to CIT

Insulin infusion was adjusted to maintain a target blood glucose concentration of 4.4–6.1 mmol/L (80–110 mg/dL) in the IIT group and 10–11.1 mmol/L (180–200 mg/dL) in the CIT group, as described previously [14]. 

### 2.3. Human MCP-1, sICAM-1, and TF Analysis 

Human MCP-1, sICAM-1, and TF were analyzed in EDTA containing samples using a Duoset ELISA (R&D System, Minneapolis, MN, USA) according to the manufacturer’s recommendations. Microplate-format colorimetric human MCP-1, sICAM-1, and TF solid phase ELISA immunoassays were established and validated. A DuoSet Ancillary Reagent Kit 2 (R&D Systems, Catalog # DY008) containing 96-well microplates, plate sealers, substrate solution, stop solution, plate coating buffer (PBS), wash buffer, and reagent diluent concentrate were used to establish the ELISA immunoassays.

### 2.4. Statistical Analysis

Data analyses were performed with the statistical package for social sciences (SPSS; release 20, SPSS Inc). Categorical data are presented as frequencies and percentages and numerical data as mean and standard deviation (SD) or medians with quartiles for skewed data. Because of a lack of interaction between the feeding and the IIT, the primary analysis was performed for IIT versus CIT and permissive underfeeding versus target feeding, as described in the original trial [14]. Secondary exploratory analyses were performed comparing the 4 groups. Baseline characteristics and biomarkers variables were compared across randomization groups (IIT vs CIT and permissive underfeeding vs target feeding) using ANOVA or the Kruskal–Wallis test for continuous variables and a chi-square test for categorical variables. A multiple linear regression model was used to assess the effect of patients’ baseline characteristics on the plasma baseline inflammatory mediators/biomarkers. The model included the log-transformation of these biomarkers (dependent variables) due to the lack of normality. The estimated coefficients of the patients’ baseline characteristics (independent variables) from the regression models were then exponentiated to obtain the estimated percent of change for these biomarkers. Linear mixed models (LMMs) were used to assess the differences in the inflammatory mediators/biomarkers across randomization groups and the time points simultaneously. Two main aspects of the mixed model were assessed prior to estimating the model parameters, the variance–covariance matrix design of the repeated measure (baseline, day 3, day 5, and day 7) and the significance of the interaction term between randomization groups and repeated measures. A test with a *P*-value < 0.05 was considered statistically significant.

## 3. Results

### 3.1. Patients’ Baseline Characteristics

The study included 91 ICU patients (IIT: 48 patients; CIT: 43 patients; permissive underfeeding: 46 patients; target feeding: 45 patients). The average age of the patients was 51.1 ± 20.8 years, with an average BMI of 28.8 ± 8.1 (kg/m^2^) and a blood sugar at baseline of 12 ± 4.6 mmol/L. One-third of the patients were females, and 36 (39.6%) were diabetic. Table 1 shows the baseline characteristics of the study groups stratified by the intervention. None of the patients’ baseline characteristics were statistically significantly different across randomization groups. There was also no significant difference in the baseline when four groups (permissive underfeeding with intensive insulin therapy (IIT) or conventional insulin therapy (CIT) and target feeding with IIT or target feeding with CIT) were compared (Appendix A).

### 3.2. Intervention

Study caloric target, average daily caloric intake, and percent caloric intake/requirement were significantly higher in target feeding groups than in permissive underfeeding groups (*P*-value < 0.01; Table 2, Appendix A). Average daily insulin doses were significantly higher and average glucose levels were lower in IIT groups than in CIT groups (*P*-value < 0.01). However, there was no difference between the insulin dose and glucose level between permissive underfeeding and target feeding (Appendix A). 

### 3.3. Outcomes

Table 2 summarizes the outcomes of the ICU patients by randomization group. Twenty-eight-day mortality, renal replacement therapy, length of stay at the hospital, and mechanical ventilation duration were not statistically different across randomization groups. Similarly, none of the outcomes showed a significant statistical difference when the four groups were compared (Appendix A).

### 3.4. Plasma Inflammatory Mediators/Biomarkers (TF, MCP-1, and sICAM-1)

Among tested predictors, only age had an association with TF (*P*-value < 0.01). For every 10-year increase in a patient’s age, TF (pg/mL) increased by 3.46% (Table 3). However, sICAM-1 was associated with SOFA at baseline and with the platelet count (*P =* 0.02 and *P =* 0.07, respectively). For each one-unit increase in SOFA, sICAM-1 increased by 8.65%, and for each 100 × 10^9^/L increase in the platelet count, sICAM-1 increased by 0.1%. None of the baseline characteristics had a significant effect on MCP-1. Figure 1 compares plasma inflammatory mediators/biomarkers (MCP-1, sICAM-1, and TF) by randomization group at the baseline, day 3, day 5, and day 7 of the ICU. MCP-1, sICAM-1, and TF were not different over time by randomization into IIT versus CIT or into permissive underfeeding versus target feeding. Appendix A shows a comparison of the inflammatory markers between four groups, which were also not different.

### 3.5. Multivariable Model

We compared two designs of the variance–covariance matrix, unstructured versus diagonal, using the likelihood ratio test (Appendix A). In all three models of inflammatory mediators/biomarkers (MCP-1, sICAM-1, and TF), the unstructured variance–covariance matrix was accepted and employed (*P-*value < 0.01). The interaction term between time and randomization was tested for the three biomarkers (MCP-1, sICAM-1, and TF), and none were significant, which suggested that the interaction term should be removed from the model and only the main effect term should be included. The model estimates are presented in Table 4. For MCP-1, there was no significant difference between randomization groups, while there was a time effect. All time points (days 3, 5, and 7) had significantly lower MCP-1 compared to the baseline (*P-*value < 0.01). For sICAM-1, neither randomization group nor time had a significant effect. However, SOFA at baseline and platelets had a significant effect (*P-*value < 0.01 for both). For TF also, there was no significant difference between the randomization groups and time, while the age of the patient had a significant effect (*P*-value < 0.01).

## 4. Discussion

In this study, we found that neither insulin therapy (ITT vs CIT) nor caloric intake (permissive underfeeding vs target enteral feeding) had a significant effect on plasma levels of MCP-1, sICAM-1, and TF in critically ill patients. 

Considering the potential association of inflammation with an activated coagulation system, we examined TF as a marker of inflammation since it is an important initiator of the coagulation system and inflammation and has been shown to be a strong prognostic marker for short-term mortality in severe sepsis and sepsis-induced acute respiratory distress syndrome patients [15]. These actions of TF may have further subtle effects on inflammation, since activated coagulation proteases may trigger proinflammatory processes. TF is expressed by various cell types [16]. Monocytes are a major source of TF in blood, and inflammation induces monocyte TF and procoagulant activity [17]. The data regarding the effect of insulin on TF is controversial. Insulin suppresses TF production by LPS-stimulated monocytes in vitro [18] and in vivo in healthy subjects [5] Additionally, insulin has been shown to suppress the proinflammatory transcription factor early growth response-1 (Egr-1), which regulates the expression of TF and plasminogen activator inhibitor-1 (PAI-1), in normal obese subjects [17]. On the other hand, in vitro addition of high glucose and high insulin increased TF procoagulant activity (TF-PCA) [3] Similarly, insulin and glucose infusion for 24 h in healthy subjects increased TF-PCA [3]. The modulatory effects of insulin or caloric intake on TF have never been studied in the ICU setting. In our results, plasma TF levels did not change at all time points compared to other randomization groups. These results add further uncertainty to the role of TF in the ICU setting and to whether hyperglycemia or hyperinsulinemia regulate its expression in the ICU setting as patients are infused with anticoagulants, considering the relative prevalence of venous thromboembolic diseases in the ICU setting. Similarly, our data did not support the notion that caloric intake modulates TF levels in the ICU setting. 

MCP-1 is a key chemokine and a potent mediator of chemotaxis, thus mediating a variety of inflammatory-promoting biological activities. A growing body of evidence has shown that high circulating levels of MCP-1 following sepsis are strongly correlated with organ dysfunction and mortality following sepsis [6,19]. While serum levels of MCP-1 have been associated with sepsis severity, high levels of seven cytokine panels, including MCP-1, have been found to predict the onset of sepsis in pre-symptomatic ICU patients [20]. Interestingly, the MCP-1 polymorphisms s1024611 and rs2857656 have been shown to be associated with sepsis susceptibility and development [21]. However, neither permissive underfeeding nor IIT or CIT showed any significant differences in the ICU setting. This lack of effect could have been the result of heterogeneous critically ill patients in our study. Additionally, the modulatory effects of insulin and caloric intake on MCP-1 levels were mainly examined in either healthy obese or noncritically ill patients with type 2 diabetes rather than in the ICU setting. 

In patients with prolonged critical illness, IIT resulted in a small but significant 15% reduction in sICAM-1 [7]. The extent of this decrease was not liable to cause a notable change in clinical outcomes, despite a correlation with improvement in endothelial function. Additionally, sICAM-1/creatinine and MCP-1/creatinine ratios were significantly decreased in patients with type 2 diabetes mellitus (T2DM) when treated with IIT [22]. In obese patients with chronic inflammation and insulin resistance, a 2-U/hr insulin infusion lowered the plasma concentrations of sICAM-1 and MCP-1 while maintaining blood glucose at normal levels [4]. Caloric intake was also shown to affect the levels of sICAM-1. In patients with chronic obstructive pulmonary disease (COPD), the concentration of sICAM-1 was inversely correlated with dietary intake [23]. Serum levels of sICAM-1 were also induced postprandially in diabetic patients [24] and in normal subjects post-glucose intake, possibly by modulating the key proinflammatory transcription factor nuclear factor κB (NF-κB) pathway [25]. Although hyperglycemia has been shown to induce sICAM-1, while insulin inhibits it, neither permissive underfeeding nor IIT showed any significant changes in serum sICAM-1 in our study. 

The lack of differences in the inflammatory mediators examined (MCP-1, sICAM-1, and TF) in our 2 × 2 factorial design randomized controlled trial could have been due to the difficulty in achieving glycemic control, as it is a complex task during an ICU stay, and to the moderate difference in caloric intake between the intervention groups. Although we achieved significant differences in calories and glucose levels, we cannot exclude that larger differences may have had an effect on these biomarkers. The severity of illness and degree of insulin resistance during an ICU stay may cause changes in nutritional delivery and other interventions (e.g., corticosteroid administration) and may produce frequent changes in insulin requirements [26]. Currently, the development of a multitude of intravenous insulin protocols in the ICU is lacking a unified optimal protocol to control hyperglycemia [27,28,29,30,31]. Additionally, targeting homogenous critically ill patients could be a better approach than the heterogeneous critically ill patients included in our study. A future study targeting larger and more homogenous critically ill patients could be a better experimental design that would minimize the high variability in the basal levels of the inflammatory mediators we observed in our study. Moreover, most of the studies examining the effects of caloric intake and insulin on inflammation have been conducted in non-ICU settings with more controlled homogenous subjects, either healthy obese or non-critically ill patients with type 2 diabetes. This could account for the result differences between these studies and our study. Additionally, there were other limitations in our study, including that the caloric requirements were estimated and not measured, and also the lower average daily protein intake administered in all the studied groups. Predictive equations to estimate a patient’s energy requirements are a quick and noninvasive useful tool, but the ability of these equations to accurately predict caloric needs has long been debated. Moreover, a hallmark of critical illness is muscle wasting related to a dramatic increase in muscle protein catabolism in the setting of inflammation. Catabolic critical illness results in a protracted and dramatic loss of nitrogen and reduces exogenous amino acid deposition into endogenous proteins. The lower average daily protein intake administered in all groups may represent a shortcoming, although this protein intake was representative of the practice of ICUs. It remains unclear at this point whether higher protein intake in the acute phase of a critical illness is beneficial or harmful, and ongoing randomized controlled trials are ongoing to address this issue.

The results of our study on these mediators were consistent with increasing data about the lack of a beneficial clinical effect of IIT and increasing caloric intake on the outcome of critically ill patients [32,33,34,35,36,37,38,39,40]. The 2 × 2 factorial design randomized controlled trial (RCT) showed no difference in the primary endpoint of 28-day mortality, although hospital mortality, which was a secondary endpoint, was lower with permissive underfeeding. However, a larger multicenter RCT by our group showed no differences in mortality between permissive underfeeding and target feeding. Similarly, insulin therapy in ICU settings has been a subject up for debate. These recent results dampen the routine use of intravenous insulin therapy in critically ill patients. In conclusion, neither insulin therapy (ITT vs CIT) nor caloric intake (permissive underfeeding vs targeted enteral feeding) had any significant effect on the inflammatory mediators measured in critically ill patients (MCP-1, sICAM-1, and TF), despite previous studies showing modulation of these inflammatory mediators through insulin infusion and caloric intake.

## Figures and Tables

**Figure 1 nutrients-11-00987-f001:**
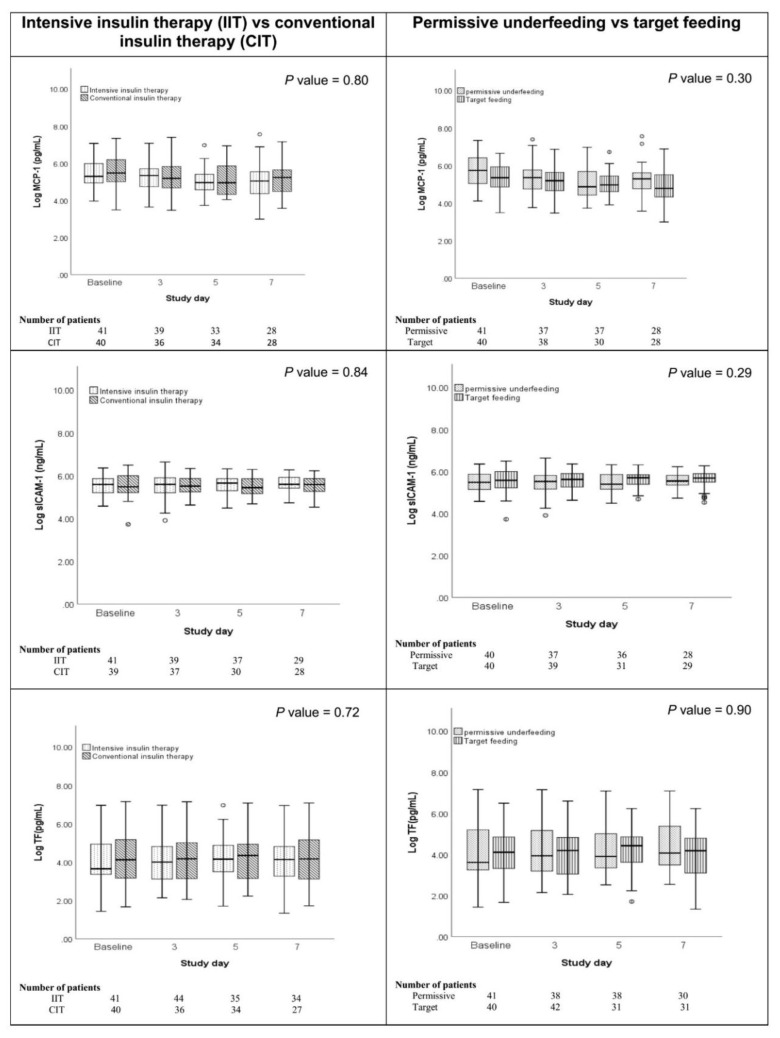
Plasma TF, MCP-1 and sICAM-1 by randomization group at each point of time expressed as box plots. Results are presented as Log.

**Table 1 nutrients-11-00987-t001:** Baseline characteristics of all patients and by randomization group.

Variable	All Patients	Intensive Insulin Therapy (*n* = 48)	Conventional Insulin Therapy (*n* = 43)	*P*-Value	Permissive Underfeeding (*n* = 46)	Target Feeding (*n* = 45)	*P*-Value
Age (years), mean ± SD	51.1 ± 20.8	51.1 ± 19.8	51.1 ± 22	1.00	50.5 ± 20.4	51.8 ± 21.3	0.77
Height (cm), mean ± SD	164 ± 11.1	163.9 ± 11.8	164.2 ± 10.5	0.91	165.6 ± 10.2	162.3 ± 12	0.18
Weight (kg), mean ± SD	76.9 ± 19.2	76.1 ± 19.8	77.8 ± 18.8	0.67	79.2 ± 17.7	74.5 ± 20.6	0.25
Body mass index, mean ± SD	28.8 ± 8.1	28.5 ± 8.1	29.1 ± 8.1	0.71	29.2 ± 7.9	28.4 ± 8.4	0.66
Inclusion of blood sugar at baseline (mmol/L), mean ± SD	12 ± 4.6	11.7 ± 4.4	12.3 ± 4.7	0.59	12.3 ± 4.8	11.6 ± 4.3	0.49
APACHE II, mean ± SD	25.4 ± 7.8	26.2 ± 7.6	24.5 ± 8	0.31	24.8 ± 7.4	26 ± 8.2	0.46
SOFA-Day1, mean ± SD	10.1 ± 3	10.3 ± 3	10 ± 3	0.62	9.8 ± 2.9	10.5 ± 3.1	0.28
Creatinine (µmol/L), mean ± SD	158.1 ± 157.3	166.3 ± 175	149.1 ± 136.3	0.61	126.4 ± 76.4	190.6 ± 206.1	0.06
Platelets 10^9^/L, mean ± SD	221.7 ± 148.3	208.1 ± 135.1	237.2 ± 162.4	0.36	216 ± 151	227.4 ± 147	0.72
INR, mean ± SD	1.4 ± 0.6	1.5 ± 0.8	1.3 ± 0.3	0.09	1.4 ± 0.7	1.3 ± 0.4	0.29
PaO_2_:FIO_2_, mean ± SD	200.6 ± 110.7	214.4 ± 114.7	185.1 ± 105.2	0.21	197 ± 120.2	204.2 ± 101.2	0.76
GCS, mean ± SD	7.7 ± 3.6	7.1 ± 3	8.2 ± 4.2	0.16	7.4 ± 3.4	7.9 ± 3.8	0.50
Gender (female), no. (%)	30 (33)	18 (37.5)	12 (27.9)	0.33	13 (28.3)	17 (37.8)	0.33
Diabetes, no. (%)	36 (39.6)	19 (39.6)	17 (39.5)	1.00	20 (43.5)	16 (35.6)	0.44
Vasopressor, no. (%)	24 (26.4)	29 (60.4)	28 (65.1)	0.64	30 (65.2)	27 (60)	0.61
Sepsis, no. (%)		13 (27.1)	11 (25.6)	0.87	12 (26.1)	12 (26.7)	0.95
Admission category, no. (%)							
Medical	56 (61.5)	30 (62.5)	26 (60.5)	0.97	25 (54.3)	31 (68.9)	0.36
Non-operative trauma	18 (19.8)	9 (18.8)	9 (20.9)		11 (23.9)	7 (15.6)	
Post-operative	17 (18.7)	9 (18.8)	8 (18.6)		10 (21.7)	7 (15.6)	

*P*-value calculated using ANOVA, Kruskal–Wallis or chi-square as appropriate. BMI: Body mass index; APACHE II: Acute Physiology and Chronic Health Evaluation II; SOFA: Sequential Organ Failure Assessment; INR: International normalized ratio; PaO_2_:FIO_2_ ratio: The ratio of partial pressure of oxygen to the fraction of inspired oxygen; GCS: Glasgow Coma Scale; ICU: Intensive care unit; SD: Standard deviation. To convert to conventional units in mg/dL, divide by 0.0555 for glucose, 88.4 for creatinine, and 17.1 for bilirubin.

**Table 2 nutrients-11-00987-t002:** Caloric intake, protein intake, insulin, and glucose and outcome data in the permissive underfeeding and target feeding groups and in the intensive insulin therapy and the conventional insulin therapy groups.

Variable	Intensive Insulin Therapy (*n* = 48)	Conventional Insulin Therapy (*n* = 43)	*P*-Value	Permissive Underfeeding (*n* = 46)	Target Feeding (*n* = 45)	*P*-Value
Calculated caloric requirement (kcal/day), mean ± SD	1747 ± 282	1816 ± 333	0.29	1840 ± 324	1718 ± 280	0.06
Study caloric target (kcal/day), mean ± SD	1474 ± 303	1510 ± 372	0.61	1322 ± 227	1664 ± 343	<0.01
Average daily caloric intake (kcal/day), mean ± SD	1207.3 ± 362.2	1276.7 ± 376.2	0.37	1108.0 ± 269.2	1375.1 ± 408.3	<0.01
Percent caloric intake/requirement (%), mean ± SD	69.6 ± 20	71.35 ± 18.8	0.68	61.1 ± 13.5	80 ± 19.8	<0.01
Calculated protein requirement (g/day), mean ± SD	72.6 ± 17.5	76.9 ± 14.1	0.21	78.2 ± 17.9	71 ± 13.2	0.03
Average daily protein intake (g/day), mean ± SD	46.7 ± 19.3	53.3 ± 16.2	0.08	51.2 ± 18.5	48.4 ± 17.9	0.47
Average enteral calories, mean ± SD	1030.5 ± 418.1	1125.7 ± 346.1	0.24	955.6 ± 300.6	1198.1 ± 427.7	<0.01
Average daily propofol calories (kcal)	48.6 ± 93.6	71.6 ± 130.3	0.33	52.5 ± 88.5	66.6 ± 133.2	0.56
Average daily dextrose calories (kcal), mean ± SD	128.5 ± 139.8	79.4 ± 96.8	0.06	100.3 ± 105.6	110.4 ± 140.1	0.70
Average daily insulin dose (Units), mean ± SD	67.9 ± 46.5	29.2 ± 44.3	<0.01	52.1 ± 53.1	47.0 ± 45.3	0.63
Average glucose levels (mmol/L), mean ± SD	6.4 ± 0.9	8.9 ± 1.8	<0.01	7.6 ± 2.0	7.6 ± 1.9	1.00
28-day mortality, no. (%)	6 (12.5)	12 (27.9)	0.12	9 (19.6)	9 (20)	0.96
180-day mortality, no. (%)	14 (29.2)	16 (38.1)	0.37	16 (35.6)	14 (31.1)	0.66
ICU mortality, no. (%)	7 (14.6)	11 (25.6)	0.19	9 (19.6)	9 (20)	0.96
Hospital mortality, no. (%)	15 (31.3)	16 (37.2)	0.55	16 (34.8)	15 (33.3)	0.88
Renal replacement therapy, no. (%)	8 (16.7)	4 (9.3)	0.30	7 (15.2)	5 (11.1)	0.56
Hypoglycemic episodes *, no. (%)	2 (4.2)	1 (2.3)	0.62	1 (2.2)	2 (4.4)	0.54
Hospital LOS (day), mean ± SD	93 ± 105	83 ± 87	0.61	86 ± 101	91 ± 93	0.785
ICU LOS (day), mean ± SD	15 ± 9.9	15 ± 10.7	0.99	13.3 ± 7.3	16.7 ± 12.4	0.121
Mechanical ventilation duration (day), mean ± SD	13 ± 9	14 ± 11	0.63	12 ± 7	15 ± 12	0.145

*P-*value calculated using ANOVA or Kruskal–Wallis as appropriate. IIT: Intensive insulin therapy; CIT: Conventional insulin therapy; LOS: Length of stay; SD: Standard deviation. * Hypoglycemic episode was defined as a blood glucose concentration < 2.2 mmol/L or 40 mg/dL.

**Table 3 nutrients-11-00987-t003:** Predictors of the biomarkers (TF, MCP-1, and ICAM-1) at baseline using a multiple linear regression model.

	MCP-1 (pg/mL)		sICAM-1 (ng/mL)		TF (pg/mL)	
	*P*-Value	% Change	*P*-Value	% Change	*P*-Value	% Change
Age (per 10 years)	0.45	0.60	0.30	0.50	<0.01	3.46
BMI (per 1 unit)	0.89	0.20	0.56	0.50	0.58	1.11
Inclusion of blood sugar at baseline (per 1 mmol/L)	0.51	−1.98	0.64	−0.80	0.50	2.74
APACHE II (per 1 unit)	0.43	1.82	0.55	0.80	0.80	0.80
SOFA day 1 (per 1 unit)	0.89	0.80	0.02	8.65	0.81	−1.88
Creatinine (per 100 µmol/L)	0.55	−0.05	0.29	−0.10	0.39	0.10
Platelets (per 100 × 10^9^/L)	0.58	−0.05	0.07	0.10	0.67	0.05
INR (per 1 unit)	0.50	−10.68	0.75	−3.15	0.48	17.23
PaO2:FIO2 (per 100 units)	0.33	−0.10	0.57	0.04	0.71	−0.10
GCS (per 1 unit)	0.29	−3.82	0.812	3.46	0.50	−3.34
Gender (female *)	0.68	11.96	0.28	19.48	0.84	−7.32
Diabetes (yes *)	0.82	6.72	0.62	8.55	0.46	32.45
Vasopressor (yes *)	0.60	−12.45	0.46	12.08	0.84	7.36
Sepsis (yes *)	0.34	−25.17	0.18	−21.73	0.38	−30.09
Admission category (medical vs post-operative *)	0.79	−9.43	0.78	6.40	0.94	3.98
Admission category (non-operative trauma vs post-operative *)	0.78	10.30	0.10	43.19	0.80	12.75

MCP-1: Monocyte chemoattractant protein 1; sICAM-1: Soluble intercellular adhesion molecule 1; TF: Tissue factor; * reference group. The model included the log-transformation of biomarkers. The estimated coefficients from the regression models were exponentiated to obtain the estimated percent of change in these biomarkers. Each unit of increase in the predictor corresponds to the percent of change of the biomarker.

**Table 4 nutrients-11-00987-t004:** Linear mixed models of plasma inflammatory mediators/biomarkers measured by time and randomization.

Parameter	Estimate	95% CI		*P*-Value	Parameter	Estimate	95% CI		*P*-Value	
**Dependent Variable: Log MCP-1** **(pg/mL)**										
Intercept	5.58	5.34	5.82	<0.01	Intercept	5.41	5.18	5.65	<0.01	
Intensive insulin therapy	−0.11	−0.41	0.20	0.49	Permissive underfeeding	0.22	−0.09	0.52	0.16	
Conventional insulin therapy *	0.00				Target feeding *	0.00				
Day 3	−0.23	−0.38	−0.08	<0.01	Day 3	−0.23	−0.38	−0.08	<0.01	
Day 5	−0.49	−0.67	−0.31	<0.01	Day 5	−0.49	−0.67	−0.31	<0.01	
Day 7	−0.49	−0.69	−0.29	<0.01	Day 7	−0.49	−0.68	−0.29	<0.01	
Baseline *	0.00				Baseline *	0.00				
**Dependent Variable: Log sICAM** **-1 (** **ng** **/mL)**										
Intercept	4.73	4.36	5.10	<0.01	Intercept	4.75	4.36	5.14	<0.01	
SOFA Day 1	0.06	0.03	0.09	<0.01	SOFA Day 1	0.06	0.03	0.08	<0.01	
Platelets × 10^9^	0.00	0.00	0.00	<0.01	Platelets × 10^9^	0.00	0.00	0.00	<0.01	
Intensive insulin therapy	−0.02	−0.18	0.15	0.86	Permissive underfeeding	−0.03	−0.19	0.14	0.74	
Conventional insulin therapy *	0.00				Target feeding *	0.00				
Day 3	0.04	−0.05	0.12	0.40	Day 3	0.04	−0.05	0.12	0.40	
Day 5	0.06	−0.03	0.16	0.18	Day 5	0.06	−0.03	0.16	0.18	
Day 7	0.05	-0.05	0.15	0.30	Day 7	0.05	−0.05	0.15	0.30	
Baseline *	0.00				Baseline *	0.00				
**Dependent Variable: Log TF (pg/mL)**										
Intercept	2.39	1.80	2.98	<0.01	Intercept	0.03	0.02	0.04	<0.01	
Age	0.03	0.02	0.04	<0.01	Age	2.22	1.64	2.81	<0.01	
Intensive insulin therapy	−0.04	−0.45	0.37	0.85	Permissive underfeeding	0.26	−0.14	0.67	0.20	
Conventional insulin therapy *	0.00				Target feeding *	0.00				
Day 3	0.01	−0.07	0.09	0.81	Day 3	0.01	−0.07	0.09	0.80	
Day 5	−0.05	−0.14	0.05	0.30	Day 5	−0.05	−0.14	0.05	0.30	
Day 7	0.00	−0.14	0.14	0.96	Day 7	0.00	−0.14	0.14	0.95	
Baseline *	0.00				Baseline *	0.00				

* Reference group. The interaction term was insignificant and hence was removed from the model. Unstructured variance–covariance matrix was employed.

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
