# Peer review of "Effect of Permissive Underfeeding with Intensive Insulin Therapy on MCP-1, sICAM-1, and TF in Critically Ill Patients"

_nutrients, 2019, doi:10.3390/nu11050987_

Round 1
Reviewer 1 Report
In this planned substudy of a RCT in critically ill patients, Aljada et al. investigated the impact of intensive insulin therapy and hypocaloric feeding on certain markers of inflammation/endothelial activation. They found no significant impact of both randomized interventions.
1. The study is a 2x2 RCT. However, in this substudy, 4 groups are compared (except for supplementary table 3). This reduces the statistical power to detect a difference. Moreover, the interpretation of the results is rather difficult, since the reader cannot well appreciate the overall effect of the randomized interventions.
I recommend to describe the effect of the 2 randomized interventions first (as in the original study). The comparison between the 4 groups can be kept in the online supplement.
2. For several reasons, the study likely has insufficient statistical power to detect a difference:
- Comparison of 4 groups, instead of 2X2 comparison (see comment above).
- With regard to the impact of feeding: the difference in caloric intake is small. Hence, the impact on inflammatory markers (if any) likely is small and probably difficult to detect in a relatively small study.
- With regard to the impact of intensive insulin therapy: a previous RCT has shown that the difference in sICAM1 was rather modest, yet significant in a study involving >400 patients. Since the current study is approximately 4 times smaller, the study is likely underpowered to detect such small difference.
3. Please report the number of observations at each time point (e.g. in the figures). I presume that, from day 5 on, there will be missing data due to ICU discharge and ICU mortality. How were missing data handled?
4. The study may suffer selection bias, since only one third of the original study population is included in this substudy. In fact, there are some signals that may indicate selection bias. Indeed, in the total study population, there was a trend towards decreased mortality by hypocaloric feeding. In the current study, if anything, mortality appeared to be greater in the hypocaloric feeding groups. Also the results support baseline imbalance, as MCP-1 was higher at baseline in the hypocaloric feeding groups. Please acknowledge. Moreover, please report how the study population was selected and for which reasons patients were excluded.
5. Please report the blood glucose levels (time-weighted average), as well as the insulin dose and nutritional intake on each study day per randomization group.
6. Blood glucose control during ICU stay is not well described. Please also report the time in target range, and the incidence of severe hypoglycemia in each group.
7. (introduction) ‘Thus, hyperglycemia has a pro-inflammatory effect whereas insulin has anti-inflammatory properties’. You may want to reformulate this sentence, because it would imply that studying the topic is not necessary anymore. Moreover, the study results may question this sentence.
8. With ‘renal therapy’, you probably mean renal replacement therapy. Please reformulate.
9. Table 3: Please report in the table and the results that the models examine the effect on the respective biomarker level at baseline (is now only mentioned in the methods).
10. Section 3.4: please write all observations in the past tense, to show that it is an observation and not a proven fact. (‘increased by'…, 'had effect',…)
11. ‘Supplementary table 3 and 4 compares two designs of the variance-covariance matrix’. You probably mean ‘supplementary table 4’ alone.
Author Response
Reviewer 1
Comments and Suggestions for Authors
In this planned sub study of a RCT in critically ill patients, Aljada et al. investigated the impact of intensive insulin therapy and hypocaloric feeding on certain markers of inflammation/endothelial activation. They found no significant impact of both randomized interventions.
Comment 1.
The study is a 2x2 RCT. However, in this sub study, 4 groups are compared (except for supplementary table 3). This reduces the statistical power to detect a difference. Moreover, the interpretation of the results is rather difficult, since the reader cannot well appreciate the overall effect of the randomized interventions.
I recommend to describe the effect of the 2 randomized interventions first (as in the original study). The comparison between the 4 groups can be kept in the online supplement.
Reply
Thank you for the suggestion. The analysis has been revised to 2 randomized interventions like in the original study as suggested and the 4 group analysis has been shifted to the supplement.
Comment 2.
For several reasons, the study likely has insufficient statistical power to detect a difference:
- Comparison of 4 groups, instead of 2X2 comparison (see comment above).
- With regard to the impact of feeding: the difference in caloric intake is small. Hence, the impact on inflammatory markers (if any) likely is small and probably difficult to detect in a relatively small study.
- With regard to the impact of intensive insulin therapy: a previous RCT has shown that the difference in sICAM1 was rather modest, yet significant in a study involving >400 patients. Since the current study is approximately 4 times smaller, the study is likely underpowered to detect such small difference.
Reply
- The comparison has been revised to 2 randomized intervention as described in the previous comment
- We agree that the difference in caloric intake is small between the two groups and as such the effect on the inflammatory markers will be small and difficult to detect in a small sample size. This has been addressed in the discussion section as a limitation.
- In the study involving >400 patients, IIT resulted in a small but significant 15% reduction in sICAM-1. The extent of this decrease was not liable to cause a notable change in clinical outcomes, despite the correlation with the improvement in the endothelial function. This has been discussed in the manuscript. Although the current study is approximately 4 times smaller and the study is likely underpowered to detect such small difference in sICAM-1, the sample size calculation was based on MCP-1 rather than sICAM-1. Our study measured 3 different markers.
Comment 3.
Please report the number of observations at each time point (e.g. in the figures). I presume that, from day 5 on, there will be missing data due to ICU discharge and ICU mortality. How were missing data handled?
Reply
We agree that the number of observations are reduced due to ICU discharge and mortality and as such number of patients at each time point have been added in the figure 1 as requested.
Comment 4.
The study may suffer selection bias, since only one third of the original study population is included in this substudy. In fact, there are some signals that may indicate selection bias. Indeed, in the total study population, there was a trend towards decreased mortality by hypocaloric feeding. In the current study, if anything, mortality appeared to be greater in the hypocaloric feeding groups. Also the results support baseline imbalance, as MCP-1 was higher at baseline in the hypocaloric feeding groups. Please acknowledge. Moreover, please report how the study population was selected and for which reasons patients were excluded.
Reply
We clarified in the Methods the selection of subjects as follows: Of 240 patients in the main trial, this substudy included a subset of 91 patients who were expected to stay at least 3 days and gave consent for this study. In addition, patients with ≥ 3 samples were included in the analysis.
With the revised analysis, MCP-1 levels were not different at baseline and over time.
Comment 5.
Please report the blood glucose levels (time-weighted average), as well as the insulin dose and nutritional intake on each study day per randomization group.
Reply
In response to your suggestion, following graphs have been added in the online supplement-supplement Figure 1;
Daily average blood glucose levels between IIT vs CIT and Permissive underfeeding vs target feeding
Daily average insulin levels between IIT vs CIT and Permissive underfeeding vs target feeding
Daily average caloric intake % between IIT vs CIT and Permissive underfeeding vs target feeding
Daily average protein intake % between IIT vs CIT and Permissive underfeeding vs target feeding
Comment 6.
Blood glucose control during ICU stay is not well described. Please also report the time in target range, and the incidence of severe hypoglycemia in each group.
Reply
Thank you. Hypoglycemia has been added in the Table 1 as suggested.
Glucose levels over time were added in Supplemental Figure 1.
Comment 7.
(introduction) ‘Thus, hyperglycemia has a pro-inflammatory effect whereas insulin has anti-inflammatory properties. You may want to reformulate this sentence, because it would imply that studying the topic is not necessary anymore. Moreover, the study results may question this sentence.
Reply
Thank you for the suggestion. The sentence has been reformulated:
“Thus, hyperglycemia may have a pro-inflammatory effect whereas insulin could exert anti-inflammatory properties”.
Comment 8.
With ‘renal therapy’, you probably mean renal replacement therapy. Please reformulate.
Reply
Thank you and yes “renal therapy” has been replaced with “renal replacement therapy”.
Comment 9.
Table 3: Please report in the table and the results that the models examine the effect on the respective biomarker level at baseline (is now only mentioned in the methods).
Reply
Table 3 and the results indicate that the models examine the effect on the respective biomarker level at baseline level.
Comment 10.
Section 3.4: please write all observations in the past tense, to show that it is an observation and not a proven fact. (‘increased by'…, 'had effect')
Reply
Section 3.4 has been corrected to past tense as suggested by the reviewer
Comment 11.
‘Supplementary table 3 and 4 compares two designs of the variance-covariance matrix’. You probably mean ‘supplementary table 4’ alone.
Reply
Yes, we agree that it is the supplementary table 4 that compares two designs of the variance-covariance matrix.

Reviewer 2 Report
The study was done by Aljada et al. was to study the effect of permissive underfeeding compared to target feeding and intensive insulin therapy to conventional insulin therapy on inflammatory mediators in critically ill patients. There are few comments as following:
This was a sub-study, and the main study was done and published in 2008 and 2011. The present sub-study is submitted in 2019. Were subjects of this sub-study belonging to the original study? Authors probably need to clarify this.
Previous studies (VISEP trial, Infection, 2005; NICE-SUGAR study NEJM, 2009) indicated that intensive glucose control increased total mortality compared with conventional treatment and led to more frequent episodes of hypoglycemia in the ICU. Authors need to clarify this issue and state why they conducted this study.
There is no introduction about the relationships between feeding status and inflammation.
Although this was a sub-study. Authors need to briefly state the inclusion and exclusion criteria.
Line 172 – 173: The lowest 28-day mortality was for IIT + permissive underfeeding with 4.1% compared to the highest 172 for CIT + permissive underfeeding with 36.4% (P = 0.06). The p value indicated the mortality was not significant between these two groups. Please check the p value since the percentage of mortality was very different (4.1% vs. 36.4%).
Authors measured 3 markers of inflammatory mediators, MCP-1, sICAM-1 and TF. In addition to these 3 markers, CRP level is often performed in ICU patients with inflammation. Therefore, it might be better to have CRP level in the results.
Since this was a sub-study of a previous published study, some wording should be carefully re-write in order to prevent a possible plagiarism.
Author Response
Reviewer 2
Comments and Suggestions for Authors
The study was done by Aljada et al. was to study the effect of permissive underfeeding compared to target feeding and intensive insulin therapy to conventional insulin therapy on inflammatory mediators in critically ill patients. There are few comments as following:
Comment 1
This was a sub-study, and the main study was done and published in 2008 and 2011. The present sub-study is submitted in 2019. Were subjects of this sub-study belonging to the original study? Authors probably need to clarify this.
Reply
We would like to clarify that the original trial was conducted between April 2006 and January 2008 and this sub study includes a subset of patients from the main trial and were recruited between December 2006 and December 2007. This has been made clear under the patient section as follows;
“The study found that permissive underfeeding may be associated with lower hospital mortality rate than target feeding (30.0% compared with 42.5%; relative risk: 0.71; 95% CI: 0.50, 0.99; P = 0.04) with no significant difference between IIT and CIT groups. In this a priori sub-study, we enrolled 91 consecutive patients enrolled from in the main trial between December 2006 and December 2007 and who expected to stayed at least 3 days in the ICU as judged by the treating physician and were consented for participation of in this study. Blood samples were collected in EDTA-and citrate-treated tubes on study day 1 (day 1), then on days 3, 5, and 7 of the enrollment in the trial. The samples were immediately centrifuged at 4⁰0 C for 20 minutes at 1600 x g. Plasma samples were stored at -80⁰0 C until assayed. Patients with ≥3 samples were included in the analyses.”
Comment 2
Previous studies (VISEP trial, Infection, 2005; NICE-SUGAR study NEJM, 2009) indicated that intensive glucose control increased total mortality compared with conventional treatment and led to more frequent episodes of hypoglycemia in the ICU. Authors need to clarify this issue and state why they conducted this study.
Reply
Our study preceded NICE-SUGAR. The combination of IIT and calories was not addressed in other studies to our knowledge.
Comment 3
There is no introduction about the relationships between feeding status and inflammation.
Reply
The introduction has been modified and a paragraph regarding the feeding status and inflammation has been added to reflect this relationship as follows.
“Macronutrient intake has been demonstrated to exert proinflammatory response. Pro-inflammatory transcription factors including nuclear factor kB (NF-kB), activator protein-1 (AP-1) and early growth response 1 (Egr-1) were activated by glucose intake with an increase in the expression of TNFα, matrix metalloproteinases 2 (MMP-2), MMP-9 and TF. The activation of NF-kB has also been shown following an intake of mixed meal. Thus, the intake of glucose leads to inflammatory stress and a prothrombotic state. A large mixed meal induced a more intense and prolonged inflammatory stress response in the obese than that in normal weight subjects”
Comment 4
Although this was a sub-study. Authors need to briefly state the inclusion and exclusion criteria.
Reply
However, we have added the inclusion criteria in the subject section in response to your request as follows;
“In this trial, patients aged 18 years with a blood glucose concentration of >6.1 mmol/L (110 mg/dL), receiving enteral feeding, and expected to stay for 48 h were randomly assigned to permissive underfeeding or target feeding groups (caloric goal: 60–70% compared with 90–100% of calculated requirement, respectively) and IIT or CIT (target blood glucose: 4.4–6.1 compared with 10–11.1 mmol/L, respectively)”.
Comment 5
Line 172 – 173: The lowest 28-day mortality was for IIT + permissive underfeeding with 4.1% compared to the highest 172 for CIT + permissive underfeeding with 36.4% (P = 0.06). The p value indicated the mortality was not significant between these two groups. Please check the p value since the percentage of mortality was very different (4.1% vs. 36.4%).
Reply
The analysis has been reverted to 2x2 RCT as suggested by the reviewers. The lowest 28-day mortality was for IIT group (12.5%) compared to CIT (27.9%). We revised the sentence to indicate to significant difference.
Comment 6
Authors measured 3 markers of inflammatory mediators, MCP-1, sICAM-1 and TF. In addition to these 3 markers, CRP level is often performed in ICU patients with inflammation. Therefore, it might be better to have CRP level in the results.
Reply
There are inherent problems with CRP as a biomarker in respect of disease activity. The secretion of CRP is largely IL-6 and TNFα dependent and certain therapies such as lipid lowering agents, ACE inhibitors, ARBs, antidiabetic agents, anti-inflammatory and antiplatelet agents, vitamin E, and β-adrenoreceptor antagonists lower serum or plasma levels of CRP and thus interrupt this pathway. This may lead to a decrease in CRP, which does not necessarily reflect a decrease in disease activity.
Comment 7
Since this was a sub-study of a previous published study, some wording should be carefully re-write in order to prevent a possible plagiarism.
Reply
The manuscript has been checked and there is no self-plagiarism.

Reviewer 3 Report
Aljada et al. has performed a priori sub-study in 91 consecutive patients from a main 2x2 factorial, randomized, controlled trial that was conducted between April 2006 and January 2008 (reference 8) by the same research team, that evaluated the effect of permissive underfeeding (caloric goal: 60-70% of calculated requirement) compared with target feeding (caloric goal: 90-100% of calculated requirement) and intensive insulin therapy (IIT: target blood glucose: 4.4-6-1 mmol/L) versus conventional insulin therapy (CIT: target blood glucose: 10-11.1 mmol/L) on the outcomes of 240 critically ill patients. They eventually concluded that permissive underfeeding may be associated with lower mortality rates than target feeding.
The aim of the current sub-study is to identify the effect of permissive underfeeding and IIT to conventional insulin therapy CIT on inflammatory mediators Monocyte Chemoattractant Protein-1 (MCP-1), soluble Intercellular Adhesion Molecule-1 (sICAM-1) and Tissue Factor (TF) in those 91 critically ill patients. Blood samples were collected on study day 1, 3, 5 and 7 for measuring them. They concluded that neither insulin therapy (IIT vs. CIT) nor caloric intake (permissive underfeeding versus targeted enteral feeding) had a significant effect on the measured inflammatory mediators.
While their findings are interesting, several methodological and analytical issues are of concern:
1. The lack of differences in the inflammatory mediators examined, as the authors stated, may be due to the difficulties in achieving glycemic control in a heterogenous cohort of critically ill patients and also to the moderate difference in caloric intake among the intervention groups. However, this study presents several other limitations such as that the caloric requirements were estimated and not measured and also the lower average daily protein intake administered in all the studied groups. We agree with the investigators that this negative study on the researched mediators may support the increasing data of the lack of beneficial clinical effect of ITT and increasing caloric intake on the outcome of critically ill patients, particularly if an amount of proteins, more close to requirements, would have been given to the studied patients. The authors should comment briefly on the above mentioned issues in the manuscript discussion.
Minor comments:
1. MCP-1, sICAM-1 and TF abbreviations meaning must be displayed when first appearing in the manuscript and in table footnotes when required.
2. Please double-check reference 9.
3. References must comply with the journal requirements.
Author Response
Reviewer 3
Comments and Suggestions for Authors
Aljada et al. has performed a priori sub-study in 91 consecutive patients from a main 2x2 factorial, randomized, controlled trial that was conducted between April 2006 and January 2008 (reference 8) by the same research team, that evaluated the effect of permissive underfeeding (caloric goal: 60-70% of calculated requirement) compared with target feeding (caloric goal: 90-100% of calculated requirement) and intensive insulin therapy (IIT: target blood glucose: 4.4-6-1 mmol/L) versus conventional insulin therapy (CIT: target blood glucose: 10-11.1 mmol/L) on the outcomes of 240 critically ill patients. They eventually concluded that permissive underfeeding may be associated with lower mortality rates than target feeding.
The aim of the current sub-study is to identify the effect of permissive underfeeding and IIT to conventional insulin therapy CIT on inflammatory mediators Monocyte Chemoattractant Protein-1 (MCP-1), soluble Intercellular Adhesion Molecule-1 (sICAM-1) and Tissue Factor (TF) in those 91 critically ill patients. Blood samples were collected on study day 1, 3, 5 and 7 for measuring them. They concluded that neither insulin therapy (IIT vs. CIT) nor caloric intake (permissive underfeeding versus targeted enteral feeding) had a significant effect on the measured inflammatory mediators.
While their findings are interesting, several methodological and analytical issues are of concern:
Comment 1.
The lack of differences in the inflammatory mediators examined, as the authors stated, may be due to the difficulties in achieving glycemic control in a heterogenous cohort of critically ill patients and also to the moderate difference in caloric intake among the intervention groups. However, this study presents several other limitations such as that the caloric requirements were estimated and not measured and also the lower average daily protein intake administered in all the studied groups. We agree with the investigators that this negative study on the researched mediators may support the increasing data of the lack of beneficial clinical effect of ITT and increasing caloric intake on the outcome of critically ill patients, particularly if an amount of proteins, more close to requirements, would have been given to the studied patients. The authors should comment briefly on the above mentioned issues in the manuscript discussion.
Reply
The above mentioned issues have been described in the manuscript discussion as follows:
“Additionally, there were other limitations for our study such as that the caloric requirements were estimated and not measured and also the lower average daily protein intake administered in all the studied groups. Predictive equations to estimate a patient's energy requirements are a quick and non‐invasive useful tool when used appropriately. However, as with any tool, the skill and experience of the user will affect the quality of the result. This can lead to significant variation in energy provision. However, the reality is that equations are the most widely used method for assessing nutrition support that patients need in hospital. Moreover, a hallmark of critical illness is muscle wasting related to a dramatic increase in muscle protein catabolism in the setting of inflammation. Catabolic critical illness results in a protracted and dramatic loss of nitrogen and reduces exogenous amino acid deposition into endogenous proteins. The lower average daily protein intake administered in all groups represent a shortcoming for the study and may have affected the outcome of the study. Future studies should address this issue.”
Minor comments
Comment 1.
MCP-1, sICAM-1 and TF abbreviations meaning must be displayed when first appearing in the manuscript and in table footnotes when required.
Reply
Abbreviation meanings have been added when first appearing in the manuscript.
Comment 2.
Please double-check reference 9.
Reply
Reference 9 has been corrected.
Comment 3.
References must comply with the journal requirements.
Reply
References have been modified to comply with the journal requirements.

Round 2
Reviewer 1 Report
Thank you for addressing most of my comments. I have a few remaining remarks.
In table 2, the P values for the comparison permissive versus target feeding are not reported. It seems that one column is missing (a technical error during construction of the pdf?). In addition, the mean blood glucose concentration in the permissive feeding group is lacking.
The supplementary figure (daily caloric intake, protein intake, average glucose concentrations, and insulin doses) is lacking as well.
The previous figures 1-3 were omitted. Please include these figures in the online supplement.
Throughout the manuscript, there are several mistakes regarding the labeling of tables and figures:
- The new figure 1 is not labeled as such.
- The tables are erronously numbered (numbering starts from 9 instead of 1)
- The labeling in the legends does not correspond with the actual numbering (in the supplementary appendix, labeling starts from supplementary table 5 instead of 1)
- Page 12. The reference to the tables is wrong. Supplementary table 2 -> suppl table 1; supplementary table 3 -> table 2
Supplementary table 3 is not referenced in the main text.
Please specify the definition of hypoglycemia used (which cutoff?)
Reviewer 2 Report
No further comments and suggestion.
Reviewer 3 Report
Please, rewrite and clarify the following resubmitted manuscript paragraph:
Outcomes
Table 2 summarizes the outcome of the ICU patients by randomization groups. The 28-day mortality, renal replacement therapy, length of stays at either the hospital mechanical ventilation duration were not different across randomization groups. Similarly, none of the 304 outcomes showed significant statistical difference when the 4 groups were compared (Supplemental Table 2)